# Single-cell RNA sequencing of *Plasmodium vivax* sporozoites reveals stage- and species-specific transcriptomic signatures

**Anthony A. Ruberto**[1☯¤], **Caitlin Bourke**[2,3☯], **Amélie Vantaux**[4], **Steven P. Maher**[5], **Aaron Jex**[2,3], **Benoit Witkowski**[4], **Georges Snounou**[6], **Ivo Mueller**[1,2,3]*

**1** Department of Parasites and Insect Vectors, Institut Pasteur, Paris, France, **2** Division of Population Health and Immunity, Walter and Eliza Hall Institute of Medical Research, Parkville, Victoria, Australia, **3** Department of Medical Biology, The University of Melbourne, Parkville, Victoria, Australia, **4** Malaria Molecular Epidemiology Unit, Institut Pasteur du Cambodge, Phnom Penh, Kingdom of Cambodia, **5** Center for Tropical and Emerging Global Diseases, University of Georgia, Athens, Georgia, United States of America, **6** Commissariat à l'Énergie Atomique et aux Énergies Alternatives-Université Paris Sud 11-INSERM U1184, Immunology of Viral Infections and Autoimmune Diseases (IMVA-HB), Infectious Disease Models and Innovative Therapies (IDMIT) Department, Institut de Biologie François Jacob (IBFJ), Direction de la Recherche Fondamentale (DRF), Fontenay-aux-Roses, France

☯ These authors contributed equally to this work.
¤ Current address: Center for Tropical and Emerging Global Diseases, University of Georgia, Athens, Georgia, United States of America
* mueller@wehi.edu.au

**Data Availability Statement:** All raw sequencing data generated from P. vivax sporozoites in this study are accessible in the European Nucleotide Archive (www.ebi.ac.uk/ena/) under the Bioproject

## Abstract

### Background

*Plasmodium vivax* sporozoites reside in the salivary glands of a mosquito before infecting a human host and causing malaria. Previous transcriptome-wide studies in populations of these parasite forms were limited in their ability to elucidate cell-to-cell variation, thereby masking cellular states potentially important in understanding malaria transmission outcomes.

### Methodology/Principal findings

In this study, we performed transcription profiling on 9,947 *P. vivax* sporozoites to assess the extent to which they differ at single-cell resolution. We show that sporozoites residing in the mosquito's salivary glands exist in distinct developmental states, as defined by their transcriptomic signatures. Additionally, relative to *P. falciparum*, *P. vivax* displays overlapping and unique gene usage patterns, highlighting conserved and species-specific gene programs. Notably, distinguishing *P. vivax* from *P. falciparum* were a subset of *P. vivax* sporozoites expressing genes associated with translational regulation and repression. Finally, our comparison of single-cell transcriptomic data from *P. vivax* sporozoite and erythrocytic forms reveals gene usage patterns unique to sporozoites.

### Conclusions/Significance

In defining the transcriptomic signatures of individual *P. vivax* sporozoites, our work provides new insights into the factors driving their developmental trajectory and lays the groundwork for a more comprehensive *P. vivax* cell atlas.

ID: PRJEB42435. P. vivax blood-stage scRNA-seq data were downloaded from NCBI's Short Read Archive (Bioproject ID: PRJNA603327). P. falciparum sporozoite scRNA-seq data were obtained from: https://github.com/vhowick/pf_moz_stage_atlasScripts and supporting files are available on GitHub: https://github.com/BourkeCaitlin/Pv-Spz-singleCell Archived scripts and output files as at time of publication are available on Zenodo: http://doi.org/10.5281/zenodo.6474355

**Funding:** This work was supported by the Agence Nationale de la Recherche (https://anr.fr, ANR-17-CE13-0025 to A.A.R, G.S., I.M.), the National Health and Medical Research Council of Australia (www.nhmrc.gov.au, NHMRC; 1092789 and 1134989 to I.M.), National Institute of Allergy and Infectious Diseases (www.nih.niaid.gov, 5U19AI129392 to I.M) and a NHMRC Principal Research Fellowship (www.nhmrc.gov.au, 1155075 to I.M). The funders had no role in study design, data collection and analysis, decision to publish, or preparation of the manuscript.

**Competing interests:** The authors have declared that no competing interests exist.

## Author summary

*Plasmodium vivax* is the second most common cause of malaria worldwide. It is particularly challenging for malaria elimination as it forms both active blood-stage infections, as well as asymptomatic liver-stage infections that can persist for extended periods of time. The activation of persister forms in the liver (hypnozoites) are responsible for relapsing infections occurring weeks or months following primary infection via a mosquito bite. How *P. vivax* persists in the liver remains a major gap in understanding of this organism. It has been hypothesized that there is pre-programming of the infectious sporozoite while it is in the salivary-glands that determines if the cell's fate once in the liver is to progress towards immediate liver stage development or persist for long-periods as a hypnozoite. The aim of this study was to see if such differences were distinguishable at the transcript level in salivary-gland sporozoites. While we found significant variation amongst sporozoites, we did not find clear evidence that they are transcriptionally pre-programmed as has been suggested. Nevertheless, we highlight several intriguing patterns that appear to be *P. vivax* specific relative to non-relapsing species that cause malaria prompting further investigation.

## Introduction

Malaria remains the most significant parasitic disease of humans globally, causing an estimated 229 million infections and 409,000 deaths per year [1]. *Plasmodium* spp. are the etiological agents of malaria, and at least five species are known to infect humans [1]. *Plasmodium falciparum* and *P. vivax* are the most prevalent, and both contribute significantly to the malaria disease burden [2–4]. *Plasmodium* spp. infection in humans begins with the deposition of sporozoites into the dermis when an infected *Anopheles* mosquito takes a blood meal [5]. While sporozoites must undergo replication in the liver before mounting a blood-stage infection, *P. vivax* sporozoites can develop into either a replicating or persisting (hypnozoite) form [6]. Hypnozoites can remain in the liver for weeks, months or years before activating to undergo schizogony [7], leading to a relapsing blood-stage infection. Relapsing infections are estimated to comprise up to 90% of *P. vivax* malaria cases in some regions [8–10]. Relapse-causing hypnozoites, in addition to a high prevalence of sub-detectable and often asymptomatic blood-stage infections, severely limit efforts to eradicate *P. vivax* malaria [3,11]. Recent modelling suggests that eliminating *P. vivax* malaria is not possible without programs that specifically target and cure hypnozoite infections [12].

The factors underlying the development of *P. vivax* sporozoites into replicating schizonts or their persistence as hypnozoites and subsequent activation remain poorly defined. Key questions regarding the regulation of hypnozoite biology have focused on how it differs regionally, seasonally, and between strains. Relapse frequency varies by climate and geographical region, with temperate strains exhibiting long periods of latency and tropical ones relapsing at shorter intervals [13–15]. The observation that *P. vivax* may be able to regulate hypnozoite formation in accordance with environmental conditions feeds into a hypothesis that the developmental outcome within the liver may be pre-determined in the sporozoite [16,17]. In addition, observations in humanised rodent livers have identified sympatric *P. vivax* strains with stable differences in hypnozoite formation rates [18]. This points to genetic heterogeneity among *P. vivax* sporozoites that may play a role in defining developmental fate, consistent with the tachy- and bradysporozoites proposed by Lysenko et al. [16]. System-wide studies offer an opportunity to

find evidence for sporozoite pre-programming; however, previous analyses of *P. vivax* sporozoites have been performed using bulk-sequencing approaches [19–21] which obscure variation that might exist between individual parasites.

Single-cell RNA sequencing methods (scRNA-seq) constitute a recent advancement applicable for assessing parasite-to-parasite differences. ScRNA-seq has differentiated multiple transcriptomic states among individual *P. berghei* and *P. falciparum* sporozoites [22–25]. However, the extent to which *P. vivax* sporozoites vary at the single-cell level has not been studied. Therefore, the application of scRNA-seq technology provides an opportunity to explore heterogeneity amongst *P. vivax* sporozoites and examine the existence of distinct transcriptional signatures that may help better understand the sporozoite's developmental fate.

In this study, we analyse the transcriptomes of 9,947 *P. vivax* sporozoites captured using droplet-based scRNA-seq technology. We first cross-reference the data with sporozoite bulk microarray and RNA-seq data to show consistent transcription of known genes upregulated in sporozoites. Next, we represent the data in low dimensional space and identify sporozoites in various transcriptomic states using both clustering and pseudotime trajectory methods. Finally, we perform comparative analyses with publicly available *P. falciparum* sporozoite and *P. vivax* blood-stage scRNA-seq data [23,26] and highlight both conserved and unique gene usage patterns between sporozoites and erythrocytic forms. Overall, our work provides an important, new resource for the malaria community by offering key insights into gene usage among *P. vivax* sporozoites and the factors driving their developmental trajectory at a resolution unattainable with bulk transcriptomics.

## Results

### Processing, alignment, and pseudo-bulk assessment of *P. vivax* sporozoite scRNA-seq data

Given its high-throughput capability and prior use for sporozoites from murine-infecting *Plasmodium* species [22,25], we used the 10x Genomics' gene expression platform to profile the transcriptomes of individual *P. vivax* sporozoites. We dissected and purified sporozoites from the salivary glands of *An. dirus* mosquitoes in triplicate, with each replicate comprising sporozoites derived from mosquitoes fed on a blood-meal from a different patient isolate. Sporozoites dissected and released from mosquito salivary glands were kept in Hank's Balanced Salt Solution at 4˚C to minimise their activation. Sporozoites from each replicate were infectious to primary human hepatocytes, as indicated by their ability to generate liver forms in a 384-well microtiter plate platform [27] (S1A–S1C Fig). High-content analysis of liver-stage cultures demonstrated that the sporozoites developed into hypnozoites and schizonts at a ratio of ~4:5 to ~6:4, depending on the case (S1B Fig). Between 5000–8000 sporozoites were loaded on a 10x Genomics' Chromium controller to partition the sporozoites, lyse, and uniquely-tag transcripts. After tagging the transcripts, Illumina compatible short-read libraries were generated and sequenced (Figs 1A and S1D and S1 Table).

After aligning the sequencing data to the *P. vivax* P01 genome (S2A Fig), various metrics confirmed that the libraries were of high quality (S2B Fig). An average of 48% of reads across all replicates (638,820,734/1,340,791,021) mapped to the *P. vivax* P01 genome (Fig 1B), with the remainder mapping to *An. dirus* (32%; 437,925,660/1,340,791,021) or not mapping at all (20%; 264,044,627/1,340,791,021) (S2C and S2D Fig and S2 Table).

Until recently, the undefined nature of untranslated regions (UTRs) in the *P. vivax* transcriptome was a gap in knowledge that restricted accurate quantification of transcription. Given the 10x Genomics' scRNA-seq technology captures specifically the 3' end of transcripts [28–30], RNA with long UTRs may be sequenced and mapped to the genome but remain

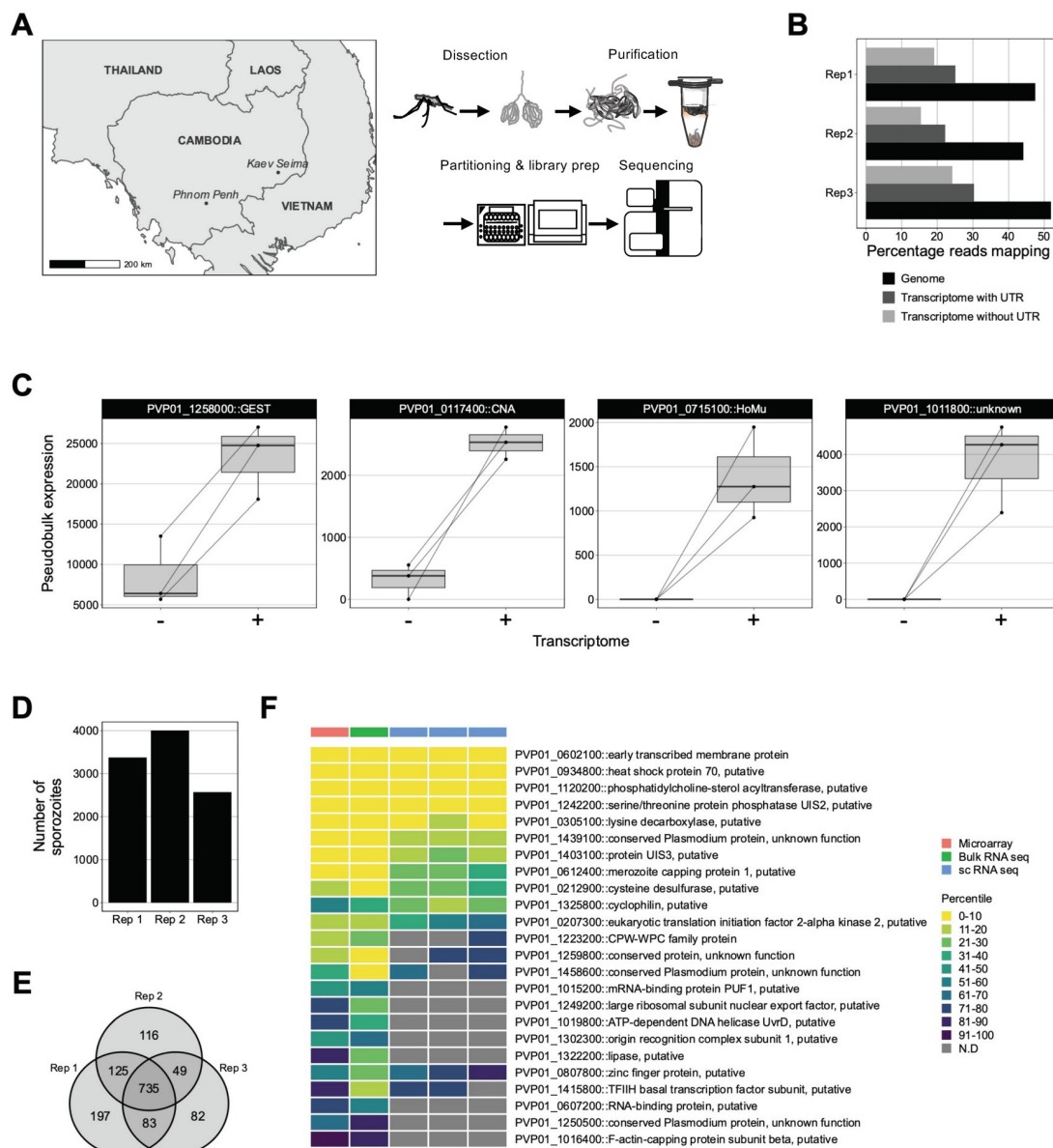

**Fig 1. Strategy used to assess *P. vivax* sporozoite transcriptomes at single-cell resolution.** (A) Schematic illustrating the geographical region, sample preparation and sequencing pipeline used to capture single-cell transcriptomes of *P. vivax* sporozoites. (B) Percentage of reads aligning to the *P. vivax* P01 genome and transcriptome (with- or without- UTR information) across the three replicates. (C) 'Pseudobulk' visualisation of UMI counts aligned to *P. vivax* P01 transcriptome without (-) or with (+) UTR information. (D) Number of sporozoite transcriptomes retained post cell- and gene- filtering. (E) Number of unique and overlapping genes detected across scRNA-seq datasets. (F) Comparison of orthologous up-regulated in infective sporozoites (UIS) genes (obtained from *P.berghei*) across three high-throughput sequencing technologies (microarray: Westenberger *et al.* 2009, bulk-RNA seq: Muller *et al.* 2019 and single-cell RNA seq; current study). To account for the differences in the total number of genes detected across technologies, gene expression values were compared using percentile ranks. *GEST*: gamete egress and sporozoite traversal; *CNA*: serine/threonine protein phosphatase 2B catalytic subunit A; *HoMu*: RNA-binding protein Musashi; UTR: untranslated region.

unquantified due to the incomplete annotation of these gene-flanking regions. This is a documented challenge for 3' capture methods in *Plasmodium and* other non-model organisms [22,25,31,32]. However, a more complete *P. vivax* transcriptome, including UTRs, was recently reported [33] and was incorporated into our alignment pipeline (S2A Fig). The inclusion of

UTR coordinates resulted in a 1.3-fold increase of reads assigned to *P. vivax* genes (Fig 1B and S3 Table). Compared to the alignment of reads using gene models lacking UTRs, UTR inclusion resulted in the detection of an additional 417 genes and an increase in counts for 1,182 genes (S3A Fig and S2 and S3 Tables). The largest increase was found for the gene encoding gamete egress and sporozoite traversal protein (PVP01_1258000) (Fig 1C and S3 Table). Other notable examples of genes with increased transcription included serine threonine protein phosphatase 2B catalytic subunit A (PVP01_0117400), RNA-binding protein Musashi (PVP01_0715100) and a conserved protein of unknown function (PVP01_1011800) (Fig 1C and S3 Table).

We next assessed the transcriptomes of 9,947 sporozoites obtained across all replicates (Fig 1D). Of the 1,387 genes detected, 735 (53%) were detected in all three replicates (Fig 1E). We observed high correlation in the transcription of genes detected in each replicate (mean Pearson correlation coefficient, R = 0.94, $p < 0.05$; S3B Fig). Genes encoding for circumsporozoite protein (PVP01_0835600), gamete egress and sporozoite traversal protein (*gest*; PVP01_1258000), and sporozoite protein essential for cell traversal (*spect*; PVP01_1212300) were among those with the highest transcription (S4 Table). Comparative analyses of our data with bulk transcriptomic studies [19,20] revealed consistent detection of various genes implicated in sporozoite biology (Fig 1F and S5 Table). These results paint a clearer picture of gene usage in *P. vivax* sporozoites by incorporating each genes' UTRs and highlight the capacity of scRNA-seq in assessing transcription in *P. vivax* sporozoites.

## Assessment of *P. vivax* sporozoites at single-cell resolution reveals transcriptomic heterogeneity

Our assessment of various per-cell metrics revealed differences in *P. vivax* sporozoites at the transcript level, specifically in the distribution of unique molecular identifiers (UMIs); the absolute number of transcripts [34], and genes detected in individual sporozoites across replicates (Fig 2A). Using Uniform Manifold Approximation Projection (UMAP) of the data to assess the transcriptomic differences visually, we found three distinct populations of sporozoites (Fig 2B). Overlaying the UMI and genes detected data on the UMAP revealed sporozoites in two main transcriptomic states: forms with higher gene usage (represented by the cells on the left side of the UMAP) and forms with lower gene usage (on the right) (Fig 2C). We first visualised transcription of well-described sporozoite membrane-associated and cell traversal proteins (Fig 2D) [35–41]. Next, given that *P. vivax* parasites can form both replicating schizonts and hypnozoites in the liver, we assessed the transcription profiles of genes implicated in sporozoite developmental fating [19]. However, no clear pattern was observed, and sporozoites in both populations transcribed genes encoding for membrane proteins, cell traversal and initiation of invasion, and translational repression (Fig 2D).

To better understand the biological significance of these populations, we used an unsupervised graph-based clustering algorithm [42] with a conservative grouping parameter (S4A Fig) to systematically divide the sporozoites into three transcriptional clusters (Fig 3A), consisting of 86 (cluster C1), 6,982 (cluster C2) and 2,879 (cluster C3) sporozoites, respectively. The number of sporozoites in each cluster varied across replicates, indicating their variability across different mosquito infections (Fig 3B). We next identified markers that define the sporozoites in each cluster using the FindAllMarkers function in Seurat [43]. We defined a marker as a gene detected in over 30% of cells in a given cluster and displaying significantly greater transcription than the other clusters. In total, we found 159 markers (adjusted p-value $< 0.05$, S6 Table). Notably, sporozoite-specific protein, S10 (PVP01_0304200), was a marker for sporozoites in C1 (Fig 3C and S6 Table), and has previously been shown to be highly transcribed in midgut

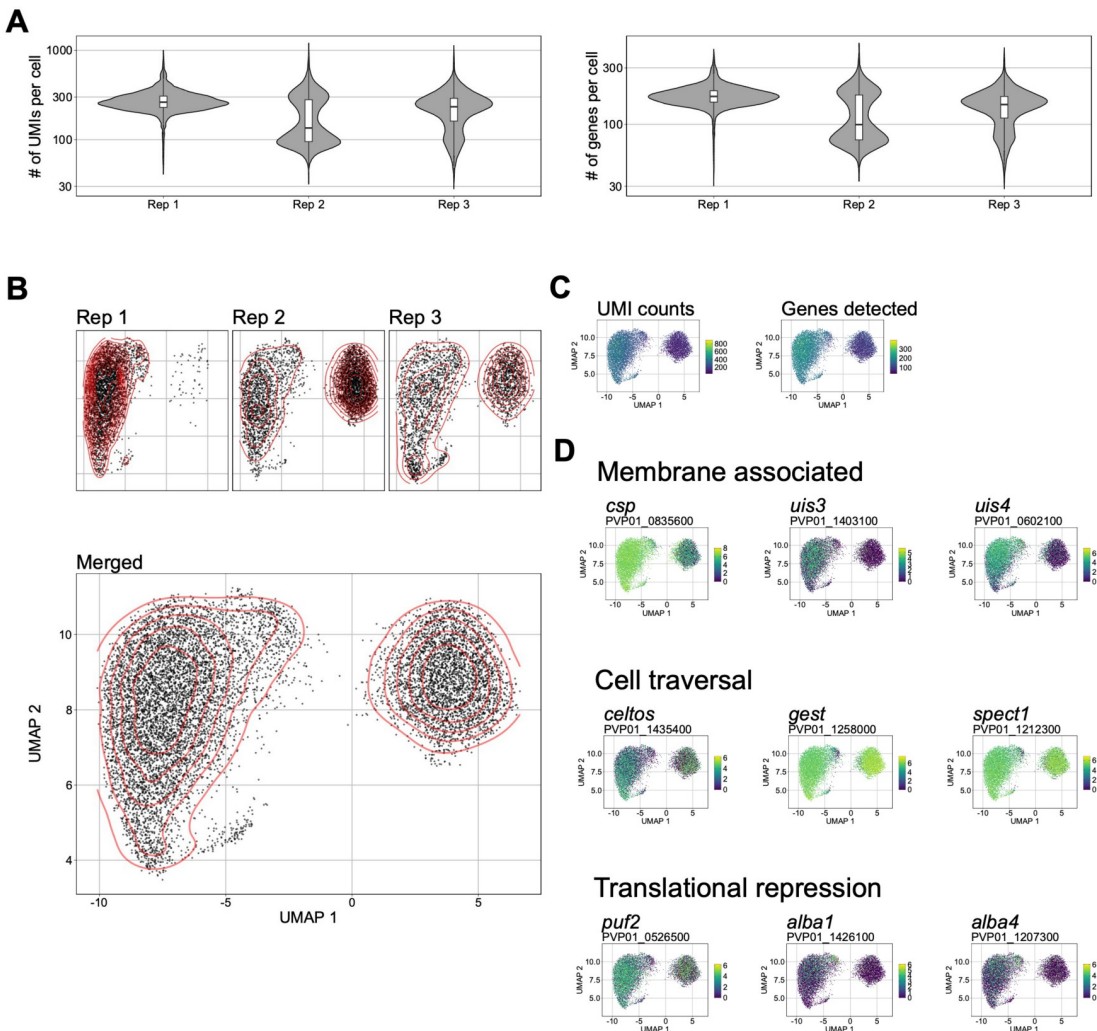

**Fig 2. Analysis of *P. vivax* sporozoite gene expression at single-cell resolution.** (A) Violin plots (with boxplox overlaid) showing the distribution of unique molecular identifiers (UMIs) per cell (left), and genes detected be cell (right). (B) UMAPs of *P. vivax* sporozoite transcriptomes, visualized per replicate (upper) and combined (lower). Red-lines represent the density of cells represented in low dimensional space. (C) UMAPs of *P. vivax* transcriptomes colored by total number of UMIs (left) and total number of genes detected (right) per cell. (D) UMAPs of *P. vivax* sporozoite transcriptomes. Cells colored by expression of various genes implicated in sporozoite biology. Scale bar: normalised expression. *csp*: circumsporozoite protein; *uis3*: upregulated-in-infective sporozoites 3; *uis4*: upregulated-in-infective sporozoites 4; *celtos*: cell-traversal protein for ookinetes and sporozoites; *gest*: gamete egress and sporozoite traversal; *spect1*: sporozoite microneme protein essential for cell traversal 1; *puf2*: mRNA-binding protein PUF2; *alba1*: DNA/RNA binding protein Alba1; *alba4*: DNA/RNA binding protein alba4.

sporozoites [22,25]. C2 markers included circumsporozoite protein (PVP01_0835600); early transcribed membrane protein (PVP01_0602100); TRAP-like protein (PVP01_1132600); and sporozoite surface protein essential for liver stage development (PVP01_0938800) (Fig 3C and S6 Table). Relative to C1 and C2, C3 had the fewest markers (20) (adjusted p-value < 0.05), and of the markers identified, their changes in transcription were modest (average $\log_2$- fold-change range 0.28–0.84; S4B Fig). Markers in this cluster included genes involved in proton transport (PVP01_0317600 and PVP01_1117400); redox response (PVP01_0835700 and PVP01_1249700); and heat shock proteins (PVP01_1440500 and PVP01_1011500) (Fig 3C and S6 Table).

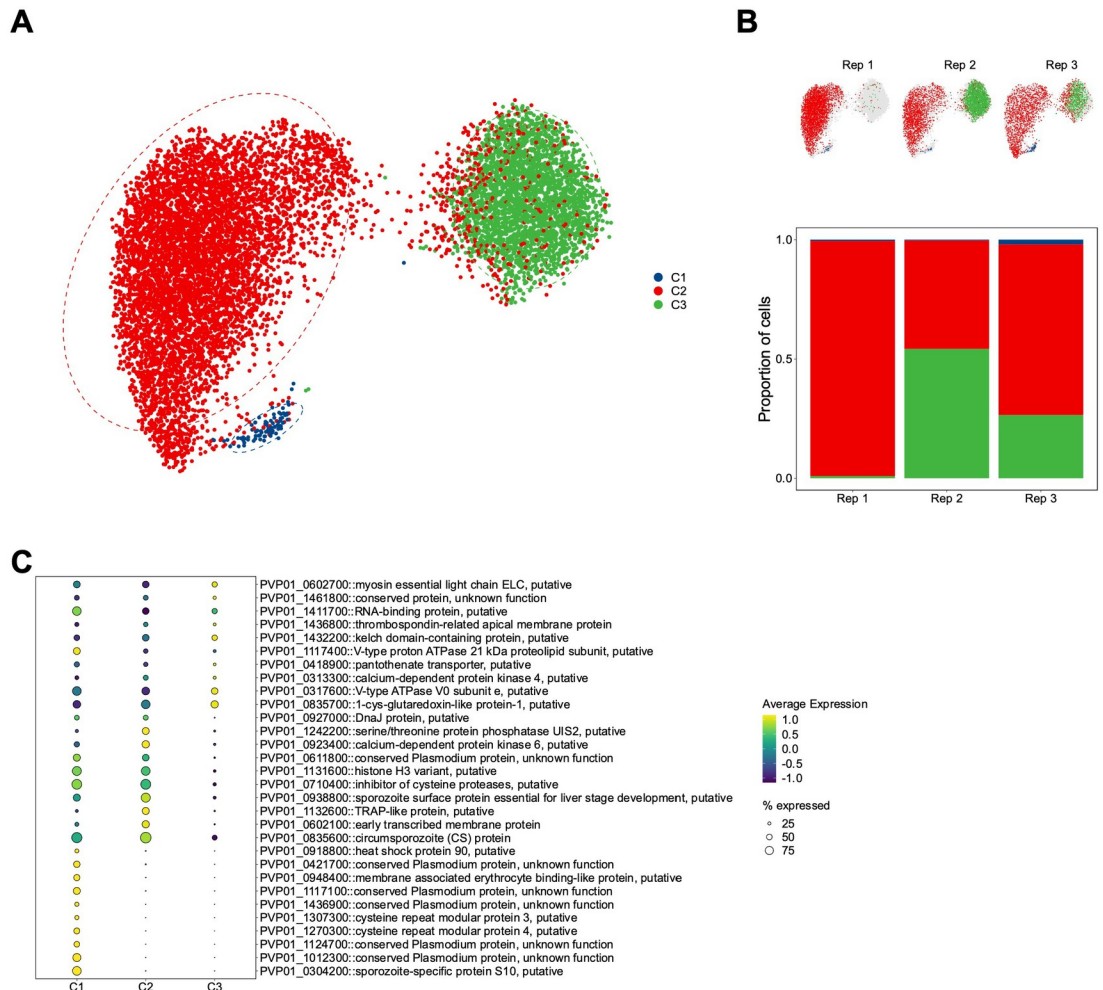

**Fig 3. Clustering and differential gene expression analysis of *P. vivax* sporozoites.** (A) UMAP of integrated *P. vivax* sporozoite transcriptomes coloured by cluster (Leiden algorithm; resolution parameter = 0.1). (B) UMAP of integrated *P. vivax* sporozoite transcriptomes split by replicate (upper) and the percentage of sporozoites in each cluster from the replicate (lower). (C) Dot plot showing top markers that distinguish each of the three clusters. The size of the dot corresponds to the percentage of sporozoites expressing the gene. Scale bar: normalised expression, scaled.

## Trajectory-based pseudotime analysis reveals various transcription patterns in sporozoites

A caveat of using a cluster-based classification method is that cells are forced into groups. In developing systems, such as sporozoites, cell transitions may be occurring more continuously. The modest number of markers defining each cluster (S6 Table) suggested that a continuum of transcriptional states may exist. We thus examined the transcriptional profiles of *P. vivax* sporozoites in the context of pseudotime. We used Slingshot [44] to construct a trajectory through the cells (Fig 4A). We observed differences in the distribution of sporozoites among replicates over pseudotime. Sporozoites from replicate 1 were unimodal and primarily enriched earlier in the trajectory, whereas sporozoites from replicates 2 and 3 were bimodal, with more cells near the end of the trajectory (Fig 4B).

Next, to assess changes in transcription as cells progressed along the trajectory, we modelled the transcription of each gene as a function of pseudotime [45]. We identified 1072

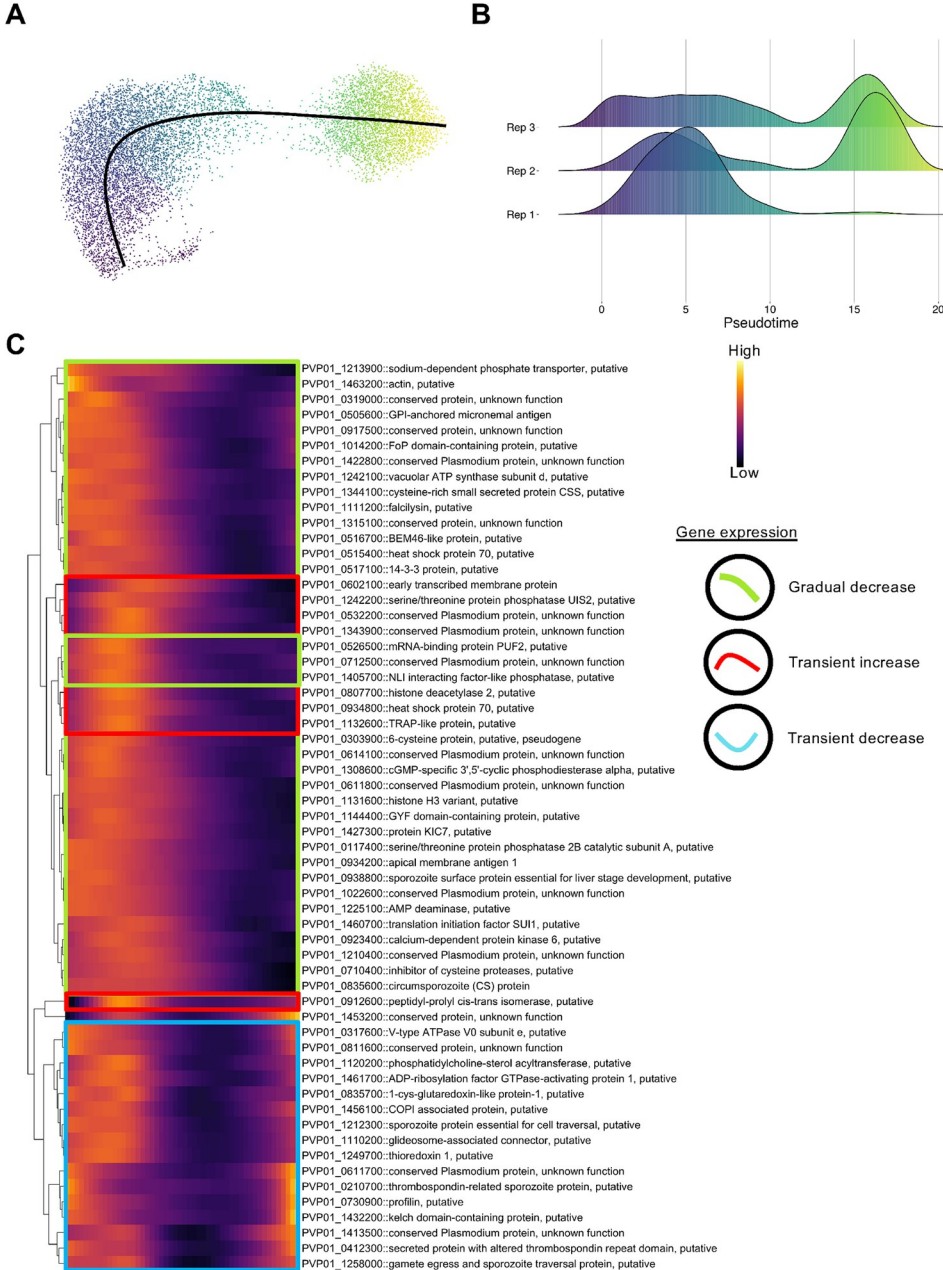

**Fig 4. Pseudotime analysis of *P. vivax* sporozoites.** (A) UMAP *P. vivax* sporozoite transcriptomes coloured by progression along pseudotime. Colour scale matches pseudotime in panel B; where darker cells are representative of cells placed earlier in pseudotime and lighter cells are projected later along the pseudotime trajectory. (B) Distribution of cells along pseudotime faceted by replicate. (C) Heatmap showing genes differentially expressed across pseudotime (Data shown: mean pseudobulk >2000, waldStat >200 and FDR < 0.01). Genes are hierarchically clustered based on their pseudotime expression profile. Scale bar: normalised expression.

differentially transcribed genes (False discovery rate (FDR) < 0.01; S7 Table). Heatmap visualisation of the data revealed three common patterns of how transcription changed over pseudotime, including a gradual decrease, transient increase, and transient decrease (Fig 4C). Our assessment of *P. vivax* sporozoites over pseudotime provides further insights into gene usage, taking into account that cell transitions in these forms may occur more continuously.

### *P. vivax* sporozoites have distinct transcriptomic signatures relative to *P. falciparum* sporozoites

Whereas *P. vivax* sporozoites can form hypnozoites in hepatocytes, *P. falciparum* sporozoites cannot. Therefore, we compared sporozoite transcriptomes from these two species to assess the extent to which their gene expression profiles differ and search for signatures of potential importance for hypnozoite formation. We first identified orthologous gene groupings to allow for the cross-species comparison [46]. After subsampling the *P. vivax* data to balance the proportions of *P. falciparum* [23] and *P. vivax* cells, we then combined the two species' datasets to produce an integrated UMAP (Fig 5A top). The projection revealed that sporozoites in the *P. vivax* and *P. falciparum* data mapped close together (Fig 5A top). The projection also revealed distinct populations of *P. vivax* and *P. falciparum* cells, with the latter grouped by anatomical location in the mosquito (Fig 5A bottom). The annotated *P. falciparum* data by mosquito location and activation status allowed us to infer the transcriptomic status of *P. vivax* sporozoites. Using these annotations as a guide, we grouped the sporozoites into four clusters (Fig 5B). Cluster 1 (ORTHO_C1) represented midgut/recently invaded sporozoites; cluster 2 (ORTHO_C2), salivary gland sporozoites; and clusters 3 and 4 (ORTHO_3 and ORTHO_4), activating or activated sporozoites. More than half of the *P. vivax* sporozoite data (63%, 949/1500) clustered with the *P. falciparum* salivary gland sporozoites in ORTHO_C2 (Figs 5C and S5A). Interestingly, some *P. vivax* sporozoites were assigned to clusters ORTHO_C1, ORTHO_C3 and ORTHO_C4, shedding light on their intrinsic heterogeneity within the salivary glands of the mosquito (Figs 5C and S5A).

For each cluster, we identified genes that display greater transcription relative to the other clusters and are conserved in both species, which we denote as conserved markers. In total, we identified 12 conserved markers (adjusted p-value < 0.05; S5C Fig and S9 Table). Relative to the other clusters, only sporozoites in the midgut/recently invaded and salivary gland clusters (ORTHO_C1 and ORTHO_C2, respectively) had conserved gene signatures. We reasoned that the disproportionate number of sporozoites for each species in ORTHO_C3 and ORTHO_C4 (Figs 5C and S5A) contributed to the lack of conserved markers detected. Of the genes with known function identified in ORTHO_C1, notable examples included sporozoite invasion-associated protein 1 (PVP01_0307900 | PF3D7_0408600), important for sporozoite exit from the mosquito midgut and colonization of the salivary gland [47] and sporozoite-specific protein S10 (PVP01_0304200 | PF3D7_0404800), a marker implicated in salivary gland invasion [22,25] (S5C and S7 Figs). In ORTHO_C2, genes linked to invasion (PVP01_1132600 | PF3D7_0616500::TRAP-like protein) and liver stage development (PVP01_0938800 | PF3D7_1137800::SPELD) were among the conserved markers identified (S5C Fig and S9 Table).

To elucidate species-specific transcription patterns within each cluster, we compared transcript levels between species. Each cluster contained orthologues transcribed solely in *P. vivax* or *P. falciparum* (Figs 5D left and S5B and S10 Table). ORTHO_C4, which was composed almost entirely of *P. vivax* sporozoites, was excluded from the analysis. We thus focused our analyses on the genes detected in both species in clusters one through three (Figs 5D right and S5B and S10 Table). In total, we identified 155 differentially transcribed genes (adjusted p-value < 0.05), 44 (28%) of which had unknown function (S11 Table). In S6A Fig, we highlight the top differentially transcribed genes within each cluster. At a broad level, genes associated with ion transport (PVP01_0317600 | PF3D7_0721900, PVP01_1014700 | PF3D7_0519200, PVP01_1117400 | PF3D7_1354400, PVP01_1242100 | PF3D7_1464700) displayed greater transcription in ORTHO_C1 *P. vivax* sporozoites (adjusted p-value < 0.05, S11 Table). Alternatively, in clusters ORTHO_C2 and ORTHO_C3, various genes associated with

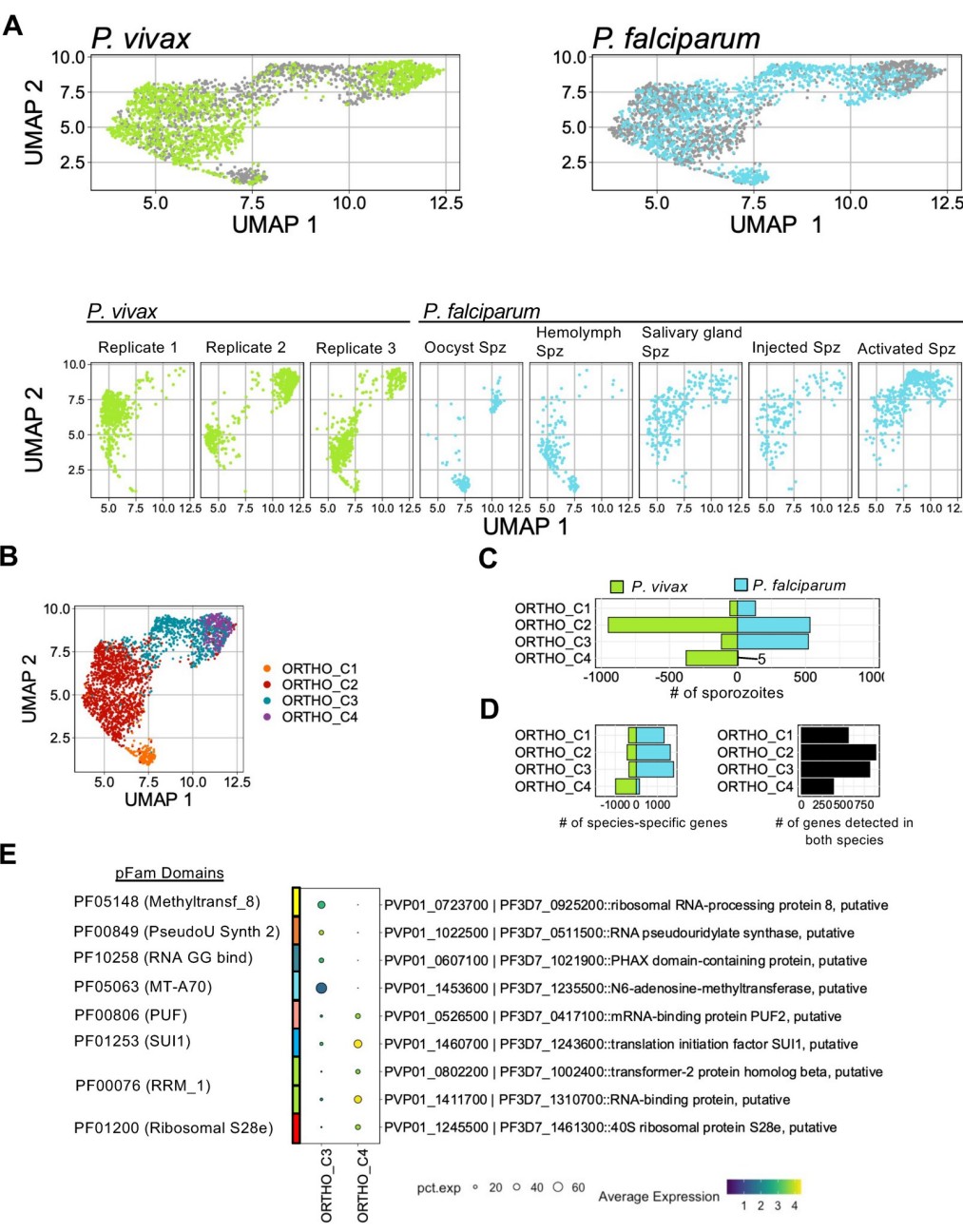

**Fig 5. Integration of *P. vivax* and *P. falciparum* sporozoite datasets.** (A) UMAPs of integrated *P. vivax* and *P. falciparum* data. Top: split and coloured by species. Bottom: split by replicate (*P.vivax*) or sporozoite status (*P. falciparum*). (B) UMAP of integrated *P. vivax* and *P. falciparum* data coloured by cluster. (C) Number of cells contributing to each cluster from *P. vivax* and *P. falciparum* samples. (D) Number of species-specific genes (left) and conserved one-to-one orthologs (right) detected in each cluster. (E) Differentially expressed genes between clusters ORTHO_C3 and ORTHO_C4 with known RNA-binding Pfam domains. Scale bar: normalised expression.

locomotion and motility (PVP01_1132600 | PF3D7_0616500, PVP01_1435400 | PF3D7_1216600, PVP01_1218700 | PF3D7_1335900) displayed greater transcription in *P. falciparum* (adjusted p-value < 0.05, S11 Table). Combined with the species-specific genes identified in each cluster, these differences highlight the distinct transcriptomic signatures between *P. vivax* and *P. falciparum*.

ORTHO_C3 and ORTHO_C4 contained sporozoites with transcriptomic signatures similar to *P. falciparum* activated sporozoites. As ORTHO_C4 was made up of primarily *P. vivax* cells, we sought to assess the extent to which sporozoites in this cluster differed from those in ORTHO_C3 containing both *P. falciparum* and *P. vivax* sporozoites. In total, we identified 98 differentially transcribed genes (adjusted p-value < 0.05, S12 Table), several (38/98; 38%) of which have no known function. Interestingly, genes that displayed significantly greater transcription in ORTHO_C4 (Fig 5E) contained Pfam domains associated with translational machinery (PF01200 and PF01253) and RNA-binding proteins (PF00076, PF00806), including mRNA-binding protein PUF2 (PVP01_0526500). PUF2 is an important eukaryotic cell-fate regulator [48], and plays a key-role in translationally repressing transcripts, including *uis3* and *uis4*, required during liver-stage development [49,50]. Interestingly, in the absence of PUF2 (Pfam: PF00806::PUF RNA binding repeat), salivary gland sporozoites may initiate exoerythrocytic development independently of the transmission-associated environmental cues [49]. These findings indicate that despite the uniform morphology and cellular invasion approach used between the species; the underlying transcriptomic signature of individual sporozoites differs between species. Whether these differences in transcription play a role in determining the sporozoite's developmental fate upon reaching the liver warrants further investigation.

## Comparative analyses of *P. vivax* sporozoite and blood-stage transcriptomes reveals conserved and stage-specific signatures

To assess gene usage across different stages of the parasite's life cycle, we compared our sporozoite data with publicly-available scRNA-seq data from *P. vivax* blood stages [26]. Before data integration, we realigned the blood-stage sequencing data to account for the UTR additions to the *P. vivax* gene models. As expected, we found a 1.4-fold gain in the number of reads mapping to gene loci when the UTRs were included in the alignment (S7A Fig). Using the cell and gene filtering pipeline established for the sporozoite scRNA-seq data, 12,469 blood-stages parasites were assessed (S7B Fig). We detected a median of 682 genes and 1,442 UMIs per cell (S7C Fig). Furthermore, low-dimensional representation of the data confirmed distinct transcriptomic signatures across each sample (S7D Fig).

After cell and gene filtering, we integrated the *P. vivax* blood-stage data with our sporozoite data (Figs 6A and S7E) and performed differential transcription analysis. We identified 208 differentially transcribed genes, 36 of which remain uncharacterised (minimum expression in 50% of cells, adjusted p-value < 0.01; S14 Table). Among the differentially transcribed genes, 59 displayed sporozoite-specific transcription (adjusted p-value < 0.01; S14 Table). As highlighted in Fig 6B, sporozoite-specific markers encoded for transcripts involved in sporozoite development, maturation, and host-cell infection [35–38,51,52]. Furthermore, we identified a small proportion of transcripts (18/208, 8%) present in a similar percentage of sporozoite and erythrocyte forms but with significantly higher transcriptional abundance in sporozoites (Fig 6C and S14 Table). Together, this integrated analysis reveals various stage-specific markers and provides the framework for creating a comprehensive reference map for *P. vivax*.

## Discussion

Our scRNA-seq data reveal transcriptional differences among *P. vivax* sporozoites at a resolution previously unattainable with bulk transcriptome-wide approaches. As is expected with droplet-based single-cell capture technologies, a comparison of our scRNA-seq data with that of bulk methods reveals a reduced overall number of genes detected using the single-cell approach [19,20,53]. However, we find that the detection of highly transcribed sporozoite genes is achieved across technologies. Herein we also describe improved approaches for processing and analysing

**A**

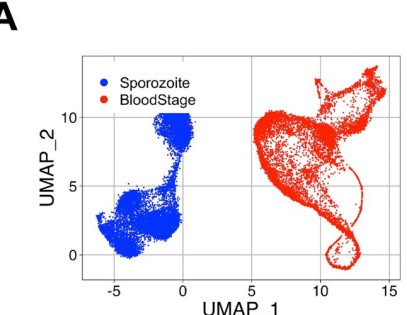

**B**

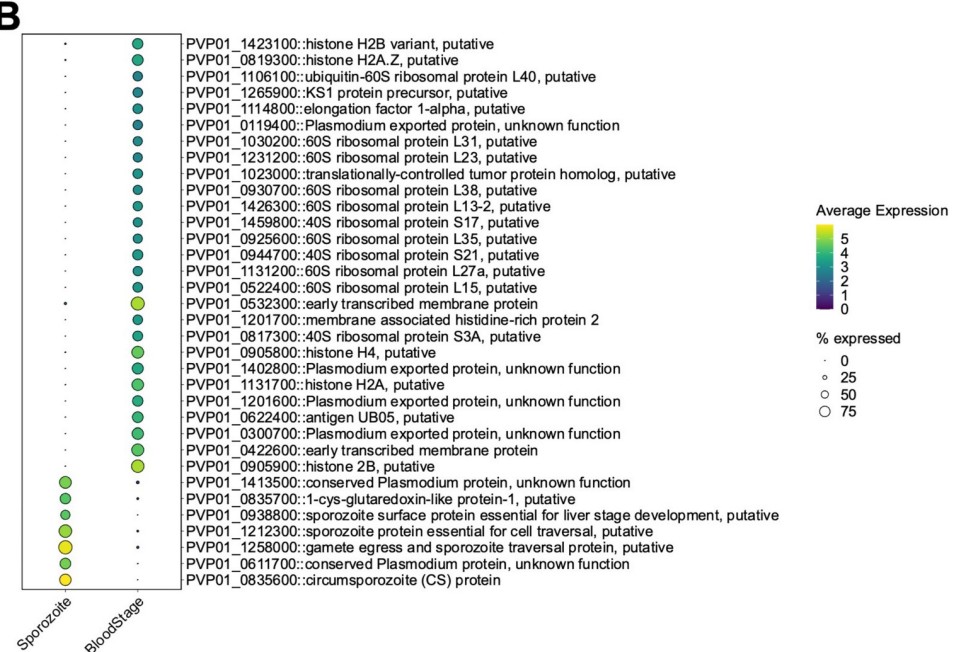

**C**

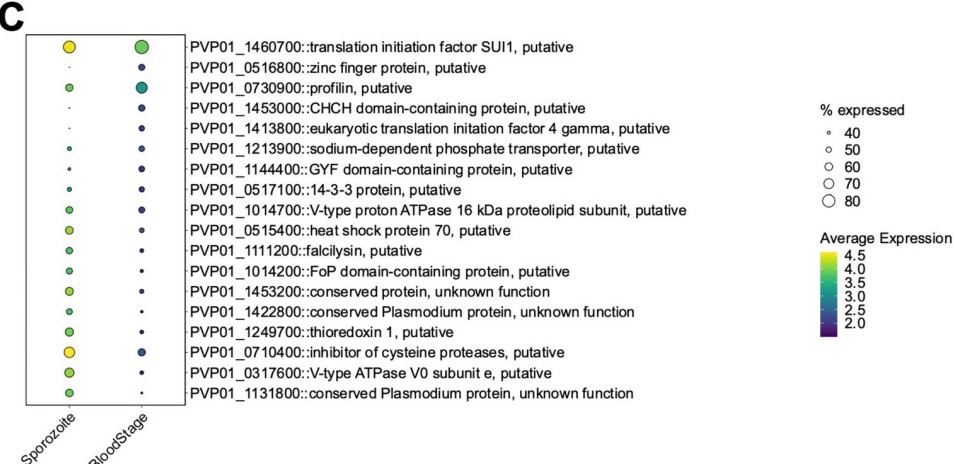

**Fig 6. Integration and comparative analyses of *P. vivax* sporozoite and blood-stage parasite transcriptomes.** (A) UMAP of integrated *P. vivax* sporozoite and blood-stage parasite transcriptomic data. (B) Dot plot highlighting the top differentially expressed genes with stage-specific specific expression patterns (absolute difference in gene detection > 65% between sporozoites and blood stages). (C) Dot plot highlighting the top differentially expressed

genes with detection in similar percentages of cells between sporozoites and blood-stage parasites. Scale bar: normalised expression.

scRNA-seq data useful for the wider community. First, our overall gene detection rate was dramatically improved by including each genes' UTRs in our alignment strategy. Second, clustering and pseudotime analyses were useful for providing new insights into transcription patterns amongst *P. vivax* sporozoites in the salivary glands of mosquitos.

Visualising the data in low-dimensional space reveals transcriptional heterogeneity amongst sporozoites. After assigning the sporozoites to clusters and performing differential transcriptional analysis, we identified annotated and unannotated genes underlying this heterogeneity. We found that some sporozoites isolated from the salivary glands show increased transcriptional activity of orthologous genes previously shown to be transcribed in *P. berghei* midgut sporozoites [22,25]. We interpret the detection of these markers in *P. vivax* to represent sporozoites that have recently invaded the salivary glands. However, we also found sporozoites that display lower gene usage, which we hypothesize is indicative of maturation. The pseudotime analysis serves to complement the cluster-based assessment of sporozoites. The ordering of sporozoites reveals a gradual decrease in the transcription of most genes. These findings likely recapitulate the transcriptional changes occurring as the sporozoite transitions from a recently invaded salivary gland sporozoite to a mature one. Translational repression of mRNA is one molecular mechanism sporozoites use to ensure timely production of proteins upon entering its host (reviewed in [54,55]). Our observation of the low number of UMIs and genes detected per cell, which corresponds with other assessments of *Plasmodium* sporozoites at single-cell resolution [22–25], suggests that transcriptional repression may also play a role in the sporozoites' preparation for host invasion.

It has been hypothesized that *P. vivax* sporozoites are pre-programmed to become replicating schizonts or hypnozoites [16,17]. However, evidence to support this hypothesis at the molecular level is lacking. We proposed that the transcriptomic analysis of individual sporozoites could test this hypothesis and might support identification of two distinct sporozoite populations, with one destined for normal hepatic development and the other for hypnozoite formation. By comparison, we expected sporozoites from *P. falciparum*, which do not form hypnozoites, would have the first but lack the second population. Our comparative analyses identify a large population of *P. vivax* and *P. falciparum* sporozoites that have conserved patterns in gene usage for various markers implicated in sporozoite biology. These include the genes encoding for sporozoite invasion-associated protein 1, TRAP-like protein, and sporozoite surface protein essential for liver stage development [47,52,56]. Interestingly, distinguishing *P. vivax* from *P. falciparum* were a subset (~25%) of *P. vivax* sporozoites transcribing genes (e.g. *puf2*) associated with translational regulation and repression. Noting the role of translational repression in eukaryotic developmental fating via RNA-binding proteins such as PUF2 [57], it is intriguing to speculate on possible links between this population and the bradysporozoites hypothesized to become hypnozoites [16].

Future analyses comparing *P. vivax* to the other sporozoites derived from parasites that cause relapsing malaria, such as *P. cynomolgi* and *P. ovale*, may help shed light on the relapsing-specific factors associated with the parasite's developmental fate. However, the differences in methodology used to generate sporozoite scRNA-seq datasets should be considered for these future studies. Of note, scRNA-seq of *P. falciparum* sporozoites was performed after dissecting salivary glands into Schneider's insect medium and then sequencing individual sporozoites following Accudenz purification, immunofluorescent staining and fluorescence-activated cell sorting [23]. In this study we used only a density gradient to purify sporozoites from salivary gland debris before loading them onto a 10x Genomics' controller to generate single-cell

RNA libraries. Both collection, purification, and capture workflows involve 1–3 hr of time from dissection to RNA capture, during which time the buffers and temperature used could impact sporozoite activation state [21,58]. Additionally, for both the *P. vivax* and *P. falciparum* datasets, pseudotime analysis was found useful for understanding the developmental state of sporozoites, meaning the day, and possibly hour, of dissection post blood-meal could impact comparisons. Despite these differences, we note the substantial overlap in *P. vivax* and *P. falciparum* developmental states in low dimensional space. Still, for future studies these parameters should be standardised as much as possible to remove potential technical artefacts from comparisons.

Previous bulk sequencing analyses comparing *P. vivax* sporozoites and blood-stage parasites have revealed distinct transcription patterns between stages [19,59,60]. Here, we corroborate these findings and show that this variation is detectable using scRNA-seq technology. Our comparison of *P.* vivax sporozoites and blood stages highlights differences in transcription that may provide insights into the factors that allow the parasite to persist in different environments. Furthermore, our integrated analysis sets the stage for future additions, with the objective of generating a comprehensive *P. vivax* single-cell atlas. Our data serves as a new resource for the malaria research community, providing a detailed assessment of *P. vivax* gene usage in sporozoites at single-cell resolution. It is indeed such knowledge that underpins the success in developing the tools and strategies that are needed to control and ultimately eliminate this globally distributed parasite species thereby freeing endemic countries from the heavy public health burden it exacts.

## Materials and methods

### Ethics statement

Research procedures and protocols for obtaining blood from patients were reviewed and approved by the Cambodian National Ethics Committee for Health Research (approval number: #113NECHR). Protocols conformed to the World Medical Association Helsinki Declaration on Ethical Principles for Medical Research Involving Human Subjects (version 2002). Informed written consent was obtained from all volunteers or legal guardians.

### Blood samples, mosquitos, and infections

Blood samples from symptomatic patients with *P. vivax* presenting to local health facilities in Mondulkiri province (Kaev Seima) in Eastern Cambodia from September to November 2019 (Fig 1A) were collected via venipuncture into heparin-containing tubes. The blood was centrifuged for 5 minutes at 3000 rpm maintained at 37˚C so that the serum be interchanged to that of a malaria-naïve individual with AB blood type (Interstate Blood Bank). Female *Anopheles dirus* colony aged 5–7 days in Mondulkiri were membrane fed, for one hour, the *P. vivax* blood-meal via custom-made insect membrane-feeders. Following the blood-meal, *An. dirus* mosquitoes were maintained at 26˚C on a 12hr:12hr light:dark cycle and fed with 10% sucrose + 0.05% PABA solution. *An. dirus* from the feed were checked for *P. vivax* oocysts at six days post-feeding via midgut dissection and 1% Mercurochrome stain. If midguts were found to contain oocysts under a 20X microscope, the remaining *An. dirus* were transported to the Institut Pasteur of Cambodia Insectary Facility in Phnom Penh, Cambodia, where they were maintained under the same conditions described above.

### Sporozoite isolations

*P. vivax* sporozoites were isolated from the salivary glands of female *An. dirus* mosquitoes 16–18 days after their infectious blood-meal, where days post blood-meal was dependent on the replicate being processed. *An.* dirus were immobilised with 70% Ethanol spray. Each replicate

represents one independent feed of a clinical isolate isolated from symptomatic *P. vivax* patients in Mondulkiri. A team of technicians performed aseptic salivary-gland dissections for a maximum period of one hour. Generally, ~75 to 100 mosquitoes were dissected in the one-hour sitting for each of the three replicates. Dissections were performed under a stereomicroscope, and salivary glands were placed in a microcentrifuge tube containing ice-cold Hanks Balanced Salt Solution. Sporozoites were then released from the salivary glands via manual disruption using a microcentrifuge pestle, and immediately purified using a discontinuous density gradient protocol adapted from Kennedy and colleagues [61], and as described previously [25]. After purification, sporozoite mixtures were diluted in HBSS to 1000 sporozoites/mL and were held on ice until further processing.

### Hepatocyte infections and liver-stage assessment

*P. vivax* hepatocyte infections were performed as previously described [27,62].

### Single-cell partitioning, library preparation, and sequencing

Approximately 5000–8000 sporozoites were loaded on a version 3- specific Chromium Chip. Chips containing sporozoite suspensions, Gel Beads in Emulsion (GEMs), and reverse transcription reagents were placed in a Chromium controller for single-cell partitioning and cellular barcoding. Barcoded cDNA libraries were generated according to the v3 Chromium Single Cell 3' gene expression protocol. cDNA libraries were loaded on individual flow cell lanes and sequenced using a HiSeq X Ten platform (Illumina) at Macrogen (Seoul, Korea). See S1 Table for sequencing statistics.

### Read alignment, cellular barcode assignment, and quantification

The *P. vivax* P01 genome (version 3, October 2020) and its corresponding general feature format (gff) file (which contained UTR coordinates) were downloaded from ftp://ftp.sanger.ac.uk/pub/genedb/releases/latest/PvivaxP01 and used to create a genome index with STAR (v2.7.3a) [63,64] with options:—runMode genomeGenerate–genomeSAindexNbases 11 –sjdbOverhang 90 (for v3 libraries), or 74 (for libraries derived from [26]). Mapping, demultiplexing and gene quantification was performed with STARsolo and the following options specified:—soloType CB_UMI_Simple—soloCBlen 16—soloUMIlen 12 (for v3 libraries) or 10 (for v2 libraries)—soloCBwhitelist /path/to/10x/version/specific/whitelist—alignIntronMin 1—alignIntronMax 2756—soloUMIfiltering MultiGeneUMI—soloFeatures Gene.

The *An. dirus* WRAIR2 genome and its corresponding GFF were downloaded from VectorBase (v49). We generated a genome index in STAR (v2.7.3a) with the same parameters as for the *P. vivax* P01 genome.

For a schematic of the alignment strategies, refer to S2A and S2C Fig. A summary of read alignment statistics for each sample with- and without- the UTR information are found in S2 and S13 Tables for the sporozoite and blood-stage data, respectively.

### Filtering and normalisation of scRNA-seq count matrices

The unfiltered (raw) matrix, features, and barcodes files generated from STAR were imported into R (version 4) [65]. We first removed rRNA encoding genes and then used emptyDrops function to distinguish between droplets containing cells and those only with ambient RNA to be discarded [66] with a lower library size limit of 40 to account for the low mRNA amounts in sporozoites (FDR < 0.001). We next removed cells with less than 60 unique genes detected in each cell library. Last, we filtered out lowly detected genes, retaining genes detected in at least

two cells with more than two unique molecular identifier (UMI) counts. Post-cell and gene filtering, the data from each replicate were normalised using Seurat's 'LogNormalize' function with the default parameters selected [43].

## Integration of *P. vivax* sporozoite scRNA-seq data

Filtered, normalised matrices from the three replicates were merged in a manner described in the Seurat (version 4) vignette, *Introduction to scRNA-seq integration*, available on the Satija Lab's website (https://satijalab.org/seurat/articles/integration_introduction.html). Briefly, highly variable features were identified in each replicate using the 'FindVariableFeatures' function with the following parameters selected: selection method = "vst", features = (genes # of detected in dataset) * 0.3. Next, integration anchors were identified using the 'FindIntegrationAnchors' function with its default settings. Lastly, using these anchors, the three datasets were integrated using the 'IntegrateData' function with its default settings.

## Dimension reduction, clustering, and cluster marker identification

Post integration, data were scaled, and dimension reduction was performed using principal component analysis (PCA) to visualise the data in low-dimensional space. Next, the UMAP dimension reduction was performed using the RunUMAP function with the parameters dims = 1:30 and umap.method = "umap-learn", n.neighbours = 20 and min.dist = 0.5. Next, an unsupervised graph-based clustering approach was used to predict cell communities. First, k-nearest neighbours were identified, and a shared nearest neighbour graph was constructed using the FindNeighbours function under the default settings. Cell communities (clusters) were then identified using the FindClusters function with the Leiden algorithm (algorithm = 4) selected. Clustering was performed at various resolutions (resolution = 0.1:1), and cluster stabilities were assessed using a Clustering tree plot [67].

To detect cluster-specific markers (differentially transcribed genes), the Seurat function FinalAllMarkers was used with the following parameters: test.use = "wilcox", min.pct = 0.3, min.diff.pct = 0.1, only.pos = TRUE, assay = "originalexp". Differentially transcribed genes were considered significant if the adjusted p value was below 0.05.

## Trajectory analysis

To infer the developmental trajectory of sporozoites, we used the Slingshot package [44] to uncover the global structure of clusters of cells and convert this structure into a smoothed lineage representing "pseudotime". Lineages were first generated using the getLineages function on the UMAP embeddings generated previously (described in *Dimension reduction, clustering and cluster marker identification*). Cluster C1 was selected as the starting cluster because it contained putative immature salivary gland sporozoite markers. Next, smoothed lineage curves were constructed using the getCurves function with the default parameters. We then used the tradeSeq package [45] to analyse transcription along the trajectory. To this end, we ran the fitGAM function on the SlingshotDataset to fit a negative binomial general additive model on the data. Based on the fitted models, the AssociationTest with default settings selected was used to test transcription changes across pseudotime. Genes were considered significantly associated with a change over pseudotime at a false discovery rate below 0.01.

## Inter-species comparison of *Plasmodium* spp. sporozoites

We grouped orthologous genes of *P. vivax P01*, *P. cynomolgi M*, *P. cynomolgi B*, *P. berghei ANKA*, *P. falciparum 3D7*, *P. yoelli yoelli 17X*, *P. chabaudi chabaudi*, *P. knowlesi H*, *P. malariae*

*UG01*, *P. ovale curtisi GH01 and Toxoplasma gondii ME49* using OrthoFinder [46]. Parameters were kept at default and gene fasta files for input were obtained from either *PlasmoDB* or *ToxoDB*, release version 51. From the 'Orthogroups' analysis, the tab-delimited output file was used to extract the species-specific gene IDs for *P. vivax* and *P. falciparum* and match these with a universal orthogroup ID (S8 Table). We used this orthogroup ID to replace rownames corresponding to *P. vivax* or *P. falciparum* gene IDs in the count matrices of *P. vivax* sporozoites (this study) and *P. falciparum* sporozoites [23]. Orthogroup IDs for merge were only retained if they had only one entry per species. In a *P. vivax*-centric approach, we retained orthogroup IDs if either *P. falciparum*, *P. berghei* ANKA, *P. cynomolgi M* or *P. cynomolgi* B had at least one corresponding gene orthologue. *P. vivax* data were processed as described earlier (section *Filtering and normalisation of scRNA-seq count matrices*) and *P. falciparum* data was obtained from https://github.com/vhowick/pf_moz_stage_atlas/tree/master/counts_and_metadata. Following the replacement of gene IDs with new orthogroup IDs, we additionally performed another round of cell filtering to account for changes in gene count per cell information with the removal of some genes without equivalent orthogroup IDs. Data integration were also performed as described earlier (section *Integration of P. vivax sporozoite scRNA-seq datasets*). The *P. vivax* data were randomly subset (500 sporozoites per replicate) for subsequent analyses to match the proportion of cells in the *P. falciparum* dataset prior to integration and account for the disproportionate number of sporozoites between the two species. Clusters of sporozoites were identified using 'FindNeighbours' function using the 'pca' reduction, dims = 1:15, k.param = 20, and 'FindClusters' function with the Leiden clustering algorithm (algorithm = 4) at a resolution of 0.2. Marker genes for the defined clusters were identified with either the 'FinderConservedMarkers' or 'FindMarkers' functions (Seurat) using a Wilcoxon Rank Sum test. The number of cells, percent difference, and fold-change parameters for each of the analyses are indicated in the R markdown document provided. Marker genes were compared to *P. vivax* bulk RNA-sequencing data from previous reports (S12 Table). Differentially expressed genes found in "hypnozoites" and "mixedLS" (adjusted p-value <0.01) from Gural *et al.* 2018 [68]; "MotilityActivation", "LiverStageEarly" and "InsectStageFinal" sporozoites from Roth *et al.* 2018 [21]; and "sporozoite" and "blood-stage" parasites from Vivax Sporozoite Consortium 2019 [19] were detailed in this comparison.

## Integration of *P. vivax* sporozoite and blood stage scRNA-seq data

Filtering and normalisation for the *P. vivax* blood-stage scRNA-seq data was performed in the same manner used for the sporozoite scRNA-seq data described in *Filtering and normalisation of scRNA-seq count matrices*. Of the 10 *P. vivax* blood-stage replicates, we used the 7 generated from samples without chloroquine treatment. Before integration, variable features for the combined sporozoite (3) and combined blood-stage (7) datasets were identified using the FindVariableFeatures function with the following parameters: selection method = "vst", features = (genes # of detected in dataset) * 0.3. Next, we used the FindIntegrationAnchors function to identify anchors between the sporozoite and blood-stage datasets. Guided by the recommendations provided by the Satija lab's vignette *Fast integration using reciprocal PCA (RPCA)* (https://satijalab.org/seurat/articles/integration_rpca.html), namely when cells in one dataset have no matching type in the other, we selected the RPCA parameter in FindIntegrationAnchors (reduction = "rpca") to identify anchors between the sporozoite and blood-stage data. Last, using these anchors, the datasets were integrated using the IntegrateData function with its default settings. Assessment of differentially transcribed genes in sporozoite and blood-stage parasites were performed in the same manner described in *Differential genes transcription analysis* with additional parameters indicated in the provided R markdown files.

## Contact for reagent and resource sharing

Further information and requests for resources and reagents should be directed to the Lead Contact, Ivo Mueller (e-mail: mueller@wehi.edu.au).

## Supporting information

**S1 Table. Illumina sequencing metrics.**
(XLSX)

**S2 Table. Alignment statistics for *P. vivax* sporozoite scRNA-seq data.**
(XLSX)

**S3 Table. Comparison of *P. vivax* sporozoite gene metrics with- and without-UTR information across each replicate.**
(XLSX)

**S4 Table. Overlap across datasets of additional genes detected in *P. vivax* sporozoites with UTR information and total counts in each replicate.**
(XLSX)

**S5 Table. Comparison of PvSpz 10x rank to Vivax Sporozoite Consortium 2019 IJP bulk RNA seq rank.**
(XLSX)

**S6 Table. Marker genes in *P. vivax* sporozoites across clusters.**
(XLSX)

**S7 Table. Genes identified as differentially expressed across pseudotime with corresponding summary statistics.**
(XLSX)

**S8 Table. Summary of orthologous genes used in cross-species integrated analysis.**
(XLSX)

**S9 Table. Conserved markers across *P. vivax* and *P. falciparum*.**
(XLSX)

**S10 Table. Overlapping and species-specific genes detected in ORTHO clusters.**
(XLSX)

**S11 Table. Genes differentially expressed between *P. vivax* and *P. falciparum* in each cluster.**
(XLSX)

**S12 Table. Genes differentially expressed between sporozoites in clusters ORTHO_C3 and ORTHO_C4.** Positive avg_log2FC: greater expression in ORTHO_C4 (*P. vivax* specific).
(XLSX)

**S13 Table. Alignment statistics for *P. vivax* blood-stage scRNA-seq data downloaded from SRA (PRJNA603327) and remapped using STARsolo.**
(XLSX)

**S14 Table. Differential gene expression between *P. vivax* sporozoites and erythrocytic-stage parasites.**
(XLSX)

**S1 Fig. Related to Fig 1. Strategy used to assess *P. vivax* sporozoite transcriptomes at single-cell resolution.** (A) Quantity of all hypnozoites, schizonts, and hepatic nuclei, as well as net growth area of schizonts, following infection of primary human hepatocytes with sporozoites from the indicated *P. vivax* replicate. Cultures were quantified at 12 days post infection. Each data point represents a single well of 24 technical replicate wells of a 384-well microtiter plate, bars represent SD across wells. The quantity of sporozoites infected into each well was 17,000 for Rep1, 16,500 for Rep2, and 18,000 for Rep3. (B) Ratio of hypnozoites versus schizonts for each isolate from 'A,' bars represent SD across 24 technical replicate wells. (C) Image of an individual culture well infected with sporozoites from the indicated isolate at 12 days post infection. Images are stitched from four fields of view taken at low-magnification (4x objective) during high- content imaging. Blue: Hoechst-stained host cell and parasite DNA, green: parasitophorous vacuole membrane detected with immunofluorescent staining with recombinant mouse anti-PvUIS4 antibody. Bar represents 500 μm. **(D)** Detailed schematic of the workflow for generating *P. vivax* single-cell RNA sequencing libraries. *P. vivax* sporozoites were manually dissected and purified by isolated the salivary glands of infected *An. dirus* mosquitoes. Sporozoites were harvested from three independent infections and three different days post- infectious blood-meal. scRNA-seq libraries were generated for the sporozoites using the 10x Genomics' 3' gene expression User Guide.
(TIFF)

**S2 Fig. Related to Fig 1. Strategy used to assess *P. vivax* sporozoite transcriptomes at single-cell resolution.** (A) Alignment of reads to *P. vivax* P01 genome, with parameters listed and output files at each stage. (B) Summary of output metrics from Illumina sequencing. (C) Alignment of reads to *An. dirus* WRAIR2 genome, with parameters listed and output files at each stage. (D) Number of reads mapping to *P. vivax* P01 genome, *An. dirus* WRAIR2 genome or unmapped to either.
(TIFF)

**S3 Fig. Related to Fig 2. Analysis of *P. vivax* sporozoite gene expression at single-cell resolution.** (A) Pairwise comparisons of transcript abundance (mean expression) with- or without- UTRs in gene models. (B) Pairwise comparisons of transcript abundance (mean expression) across the three replicates when sequencing reads are aligned to the *P. vivax* P01 genome with UTRs. Pearson's correlation coefficients (Corr, *R)* were determined using values > 0 for each pairwise comparison. ***, *p* value < 0.001.
(TIFF)

**S4 Fig. Related to Fig 3. Clustering and differential expression analysis of *P. vivax* sporozoites.** (A) Visualisation of cluster stability when resolution is increased in increments of 0.1 (start = 0.1 and end = 1.0). The size of each point is representative of the number of cells. Edges coloured by number of cells; and transparency represents the incoming node proportion (the number of samples in the edge divided by the number of samples in the node it points to). Point fill (sc3_stability) represents the calculated cluster stability. Clustering tree created with Clustree package (Zappia & Oshlack, 2018). Box indicates clustering resolution used for subsequent differential expression analysis. (B) Scatter plot of 159 genes identified as differentially expressed across the three clusters, split by the cluster. Genes displaying greater expression in each respective cluster plotted. Size of the point represents the percentage of cells expressing the gene of interest.
(TIFF)

**S5 Fig. Related to Fig 5. Integration of *P. vivax* and *P. falciparum* sporozoite datasets.** (A) Proportion of cells derived from *P. vivax* and *P. falciparum* in each cluster. (B) Number of

overlapping and unique one-to- one orthologs detected in each cluster. (C) Averaged gene expression (log1p) for genes detected in each cluster. (D) Dot plot of the top conserved genes across the two species for clusters one and two (Seurat parameters: min.pct .25, min.diff. pct = 0.125, Wilcoxon rank-sum test). Scale: Normalised expression, scaled; dot size: percentage of cells the transcript is detected.
(TIFF)

**S6 Fig. Related to Fig 5. Integration of *P. vivax* and *P. falciparum* sporozoite datasets.** (A) Dot plots of top one-to-one orthologs in each cluster that are differentially expressed between *P. vivax* and *P. falciparum*. The size of the dot corresponds to the percentage of cells expressing the gene. Scale bar: normalised expression.
(TIFF)

**S7 Fig. Related to Fig 6. Integration and comparative analyses of *P. vivax* sporozoite and blood-stage parasite transcriptomes.** Realignment, processing, and per-cell metrics of *P. vivax* blood stage 10x scRNAseq data prior to integration with the *P.vivax* sporozoite 10x scRNAseq data generated in this study. (A) Percentage of reads aligning to the *P. vivax* P01 genome (upper panel) and transcriptome (with- or without- UTR information) (lower panel). (B) Number of *P. vivax* blood-stage transcriptomes retained post cell and gene filtering. (C) Violin plots showing the distribution of genes detected per cell (upper) and the UMIs detected per cell (lower). (D) PCA plots of the samples, split by day of infection, treatment, and monkey. Cells coloured by *P. vivax* strain used during monkey infection. Samples highlighted in green are those that are integrated with the scRNA-seq sporozoite data of the current study. (E) UMAP of integrated *P. vivax* sporozoite and blood stage data coloured by sample.
(TIFF)

## Acknowledgments

We thank the *P. vivax* patients of Mondulkiri Province, Cambodia, for participating in this study. We thank the Institut Pasteur insectary staff (Makara Pring, Koeun Kaing, Nora Sambath) for *An. dirus* mosquito colony maintenance, the laboratory staff (Eakpor Piv, Chansophea Chhin, Sreyvouch Phen, Chansovandan Chhun, Sivcheng Phal, Baura Tat) for assistance with the mosquito dissections and the *in vitro* assays, and the field site manager (Saorin Kim) for logistical assistance.

## Author Contributions

**Conceptualization:** Anthony A. Ruberto, Caitlin Bourke, Amélie Vantaux, Aaron Jex, Benoit Witkowski, Georges Snounou, Ivo Mueller.

**Data curation:** Anthony A. Ruberto, Caitlin Bourke.

**Formal analysis:** Anthony A. Ruberto, Caitlin Bourke.

**Funding acquisition:** Georges Snounou, Ivo Mueller.

**Investigation:** Anthony A. Ruberto, Caitlin Bourke, Amélie Vantaux, Steven P. Maher.

**Methodology:** Anthony A. Ruberto, Caitlin Bourke, Amélie Vantaux, Steven P. Maher, Benoit Witkowski, Georges Snounou, Ivo Mueller.

**Project administration:** Ivo Mueller.

**Supervision:** Aaron Jex, Ivo Mueller.

**Visualization:** Anthony A. Ruberto, Caitlin Bourke.

**Writing – original draft:** Anthony A. Ruberto, Caitlin Bourke.

**Writing – review & editing:** Anthony A. Ruberto, Caitlin Bourke, Amélie Vantaux, Steven P. Maher, Aaron Jex, Benoit Witkowski, Georges Snounou, Ivo Mueller.

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
