## [Decision Letter · Decision Letter 0]

8 Apr 2022

Dear Ms Bourke,

Thank you very much for submitting your manuscript "Single-cell RNA sequencing of Plasmodium vivax sporozoites reveals stage- and species-specific transcriptomic signatures" for consideration at PLOS Neglected Tropical Diseases. As with all papers reviewed by the journal, your manuscript was reviewed by members of the editorial board and by several independent reviewers. The reviewers appreciated the attention to an important topic. Based on the reviews, we are likely to accept this manuscript for publication, providing that you modify the manuscript according to the review recommendations. 

Sincerely,

Paul O. Mireji, PhD

Associate Editor

Lisa Ranford-Cartwright

Deputy Editor

Reviewer's Responses to Questions

**Key Review Criteria Required for Acceptance?**

**Methods**

-Are the objectives of the study clearly articulated with a clear testable hypothesis stated?

-Is the study design appropriate to address the stated objectives?

-Is the population clearly described and appropriate for the hypothesis being tested?

-Is the sample size sufficient to ensure adequate power to address the hypothesis being tested?

-Were correct statistical analysis used to support conclusions?

-Are there concerns about ethical or regulatory requirements being met?

Reviewer #1: Yes

Reviewer #2: The objective of this study are well articulated and a sound study design, including a robust comparative analysis, is employed to address this objective. 

Despite there being no evidence of pre-programming of the infectious sporozoite while in the salivary-glands, the use of droplet-based scRNA-seq technology reveals high resolution gene usage patterns. The methods employed here set the stage for future improvements of the P. vivax single-cell atlas that can be useful in advancing the control of hypnozoite infections. 

The number of mosquitos and sporozoites used in this study has been specified and is sufficient for scRNA-seq transcriptional analysis. 

Ethic statement has been provided

A brief description of the blood samples collection protocol in line 389 should be provided.

Reviewer #3: The study is well designed with clear objectives. The sample size (5000-8000 sporozoites per chip) are sufficient for the analysis. what wasn't clear was the total number of unique parasite lines used in the initial mosquito infections and were these mixed infections or clonal. The statistical analysis were appropriate to support the conclusions. 

The authors compared silvery gland sporozoites tes single-cell transcriptimes to single cell blood stage transcriptomes of p. vivax (Sa et al., 2020) and identified similarities as well as unique features of salivary gland parasites. It would be interesting if they could also consider other data sets e.g. the Vivax Sporozoite Consortium 2019 paper on salivary glad sporozoites or the Cubi et al., 2017 paper on P. cynomolgy hypnozoites to increase the power of finding the mechanisms or signatures of commitment to maintain persistent liver infection that may occur during salivary glad stage.

**Results**

-Does the analysis presented match the analysis plan?

-Are the results clearly and completely presented?

-Are the figures (Tables, Images) of sufficient quality for clarity?

Reviewer #1: Yes

Reviewer #2: The results are clearly and completely presented with high quality and clear tables and images. All supplementary data is available.

Reviewer #3: the results were well represented

**Conclusions**

-Are the conclusions supported by the data presented?

-Are the limitations of analysis clearly described?

-Do the authors discuss how these data can be helpful to advance our understanding of the topic under study?

-Is public health relevance addressed?

Reviewer #1: Yes

Reviewer #2: The discussion has been placed into context without being overinterpreted and has answered the aims of the study.

The conclusion is supported by appropriate references and results. 

The limitations of this study, such as a standardized time of dissection post blood-meal, are not fatal but can be used to inform future research.

The closing statements can be better re-worded to capture the broad public health concern of P. vivax malaria control (IVM) beyond the molecular significance.

Reviewer #3: The work provides an opportunity for in-depth understanding P. vivax infection in mosquito stages and potential identification of key molecules and implication on the pathways that define P. vivax development including one leading to persistent liver infection.

**Editorial and Data Presentation Modifications?**

Reviewer #1: Minor revision

Reviewer #2: The legend labels for Fig S4 (A) should be increased

Reviewer #3: Minor revisions

**Summary and General Comments**

Reviewer #1: This manuscript PNTD-D-21-01772, Ruberto et al., reports on a single-cell RNA-seq transcriptome analysis of the sporozoites of P. vivax parasites obtained from infected salivary gland of mosquito, compares them with those obtained from infected liver cells. So far, although parasites in the mosquito salivary gland are essential in malaria transmission to the mammalian host, no gene expression study of P. vivax parasites in this part of the fly have been done and single cell level with only studies limited studies done on the parasite inhabiting the liver cells. In this context, the presented study has certainly an important merit toward enhancing the understanding the transmission biology of P. vivax and can lead to identification of novel drug target as well as vaccine and diagnostic candidate. Additionally, this study also contributes important literature to the malaria scientific community.

Reviewer #2: The aim of this study was to see whether the reactivation of hypnozoites in the liver responsible for relapsing infections could be attributed to pre-programming of the infectious sporozoite in the salivary-glands of the vector. The authors sought to reveal this phenomenon at transcription level in salivary-gland sporozoites. As much as there was no significant evidence of pre-programming in the salivary-gland sporozoites, this work provides a novel genomic resource for the malaria community at an impressive resolution.

Reviewer #3: This is an important piece of work giving indent analysis of salivary gland sporozoite biology and the cellular level.

PLOS authors have the option to publish the peer review history of their article (what does this mean?). If published, this will include your full peer review and any attached files.

Reviewer #1: No

Reviewer #2: Yes: Clarence M. Mang'era (Ph.D.)

Reviewer #3: No

Figure Files:

Data Requirements:

Reproducibility:

References

---

## [Decision Letter · Decision Letter 1]

4 Jul 2022

Dear Ms Bourke,

We are pleased to inform you that your manuscript 'Single-cell RNA sequencing of Plasmodium vivax sporozoites reveals stage- and species-specific transcriptomic signatures' has been provisionally accepted for publication in PLOS Neglected Tropical Diseases.

Best regards,

Paul O. Mireji, PhD

Associate Editor

Lisa Ranford-Cartwright

%CORR_ED_EDITOR_ROLE%

Paul J. Brindley

Co-Editor-in-Chief

One reviewer has noted that the process of oocyst identification in the mosquito is not clear and that maintenance of the mosquitoes should be better clarified. The reviewer has also requested for further clarification on how the mosquitoes were dissected. As handling Associate Editor, I must balance these up and consider whether these are minor observations that can be addressed without another round of peer review of the MS or if another round would be necessary. I have seriously considered these options and concluded that these are not substantive concern and can be addressed in the final/revised version of the MS.

Reviewer's Responses to Questions

**Key Review Criteria Required for Acceptance?**

**Methods**

-Are the objectives of the study clearly articulated with a clear testable hypothesis stated?

-Is the study design appropriate to address the stated objectives?

-Is the population clearly described and appropriate for the hypothesis being tested?

-Is the sample size sufficient to ensure adequate power to address the hypothesis being tested?

-Were correct statistical analysis used to support conclusions?

-Are there concerns about ethical or regulatory requirements being met?

Reviewer #1: Yes

Reviewer #2: Methods have been well described

Reviewer #3: The issues raised before have been addressed. The methods are sufficient to answer the study objectives

**Results**

-Does the analysis presented match the analysis plan?

-Are the results clearly and completely presented?

-Are the figures (Tables, Images) of sufficient quality for clarity?

Reviewer #1: Yes

Reviewer #2: Results are well presented

Reviewer #3: The result are well presented and do match the analysis plan

**Conclusions**

-Are the conclusions supported by the data presented?

-Are the limitations of analysis clearly described?

-Do the authors discuss how these data can be helpful to advance our understanding of the topic under study?

-Is public health relevance addressed?

Reviewer #1: Yes

Reviewer #2: The conclusion sufficiently describes the data presented. Limitations have been exhaustively presented in the paragraph on line 350

Reviewer #3: The conclusions are supported by data presented and where not appropriate explanations have been provided. therefore I have no further comments

**Editorial and Data Presentation Modifications?**

Reviewer #1: This revised manuscript PNTD-D-21-01772R1, Ruberto et al., reports on a single-cell RNA-seq transcriptome analysis of the sporozoites of P. vivax parasites obtained from infected salivary gland of mosquito, compares them with those obtained from infected liver cells. So far, although parasites in the mosquito salivary gland are essential in malaria transmission to the mammalian host, no gene expression study of P. vivax parasites in this part of the fly have been done and single cell level with only studies limited studies done on the parasite inhabiting the liver cells. In this context, the presented study has certainly an important merit toward enhancing the understanding the transmission biology of P. vivax and can lead to identification of novel drug target as well as vaccine and diagnostic candidate. Additionally, this study also contributes important literature to the malaria scientific community.

My previous comments were adequately addressed.

However, in pg7 line 167, in figure 2B, only two populations are visible. Please point out the three possible populations

Reviewer #2: None

Reviewer #3: none

**Summary and General Comments**

Reviewer #1: (No Response)

Reviewer #2: (No Response)

Reviewer #3: Previously raised issues have been addressed sufficiently therefore I have no further comments.

PLOS authors have the option to publish the peer review history of their article (what does this mean?). If published, this will include your full peer review and any attached files.

Reviewer #1: No

Reviewer #2: **Yes: **Dr. Mang'era Clarence M.

Reviewer #3: No

---

## [Editor Report · Acceptance letter]

1 Aug 2022

Dear Ms Bourke,

We are delighted to inform you that your manuscript, "Single-cell RNA sequencing of </i>Plasmodium vivax</i> sporozoites reveals stage- and species-specific transcriptomic signatures," has been formally accepted for publication in PLOS Neglected Tropical Diseases.

Best regards,

Shaden Kamhawi

co-Editor-in-Chief

Paul Brindley

co-Editor-in-Chief
